# A Micro-Mach–Zehnder Interferometer Temperature Sensing Design Based on a Single Mode–Coreless–Multimode–Coreless–Single Mode Fiber Cascaded Structure

**Qing Yang, Jing Tian \*, Xiao Hu, Jiajun Tian and Qiqi He**

College of Physics, Guizhou University, Guiyang 550025, China; gs.yangqing21@gzu.edu.cn (Q.Y.); gs.xiaohu22@gzu.edu.cn (X.H.); gs.jjtian23@gzu.edu.cn (J.T.); gs.qqhuo21@gzu.edu.cn (Q.H.)
\* Correspondence: jtian1@gzu.edu.cn

**Abstract:** In this paper, a temperature sensing scheme with a miniature MZI structure based on the principle of inter-mode interference is proposed. The sensing structure mainly comprises single mode–coreless–multimode–coreless–single mode fibers (SCMCSs), which have been welded together, with different core diameters. The light beam has been expanded after passing through the coreless optical fiber and is then coupled into a multimode optical fiber. Due to the light passing through the cladding and core mode of the multimode optical fiber with different optical paths, a Mach–Zehnder interferometer is formed. Moreover, due to the thermo-optic and thermal expansion effects of optical fibers, the inter-mode interference spectrum of a multimode fiber shifts when the external temperature changes. Through theoretical analysis, it is found that the change in the length of the sensing fiber during temperature detection has less of an effect on the sensitivity of the sensing structure. During the experiment, temperature changes between 20 and 100 °C are measured at sensing fiber lengths of 1.5 cm, 2.0 cm, 2.5 cm, 3.0 cm, 3.5 cm, and 4.0 cm, respectively, and the corresponding sensitivities are 65.98 pm/°C, 72.70 pm/°C, 67.75 pm/°C, 66.63 pm/°C, 74.80 pm/°C, and 72.07 pm/°C, respectively. All the corresponding correlation coefficients are above 0.9965. The experimental results indicate that in the case of a significant change in the length of the sensing fiber, the sensitivity of the sensing structure changes slightly, which is consistent with the theory that the temperature sensitivity is minimally affected by a change in the length of the sensing fiber. Therefore, the effect of the length on sensitivity in a cascade-based fiber structure is well solved. The sensing scheme has an extensive detection range, small size, good linearity, simple structure, low cost, and high sensitivity. It has a good development prospect in some detection-related application fields.

**Keywords:** temperature sensing; inter-mold interference; Mach–Zehnder interference; cascaded structure

## 1. Introduction

Optical fiber sensing technology is a current research hotspot due to its advantages such as high sensitivity, small size, easy production, strong corrosion resistance, anti-electromagnetic solid interference ability, and the practicability of optical fiber sensors in extraordinary and extreme environments. In recent years, fiber optic sensing technology has developed rapidly. Its sensing mechanism mainly lies in the particular modulation of the optical signal transmitted in the optical fiber so that a change in external physical parameters can be transformed into a change in the optical signal. Then, the change in external physical quantity can be reflected by demodulating the optical signal to detect relevant physical quantities including pressure, temperature, displacement, etc. [1–6].

Temperature sensors are widely used, and the main types of fiber optic temperature sensors are Sagnac interferometric [7–9], fiber Bragg grating [10–12], special fiber optic [13–15] and cascade sensors [16–18]. Lim et al. proposed a Sagnac interferometer, which is composed of two polarization-maintaining fibers, and its temperature sensitivity

can reach 65.3 pm/°C [19]. Fiber Bragg grating sensors were fabricated by Osman et al. to compare single-mode fiber grating and multimode fiber grating; these sensors were made with phase mask technology and had temperature sensitivities of 10.9 pm/°C and 13.23 pm/°C [20]. Special fiber optic sensors, such as those reported by Chaudhary V.S. et al., comprise twin core photonic crystal fibers for temperature sensing, and the sensitivity reaches 18.5 pm/°C in the range from 0 °C to 600 °C [21]. Ma et al. proposed a cascade-type temperature sensor based on a single mode–coreless–single mode structure with a sensitivity of 38.7 pm/°C [22]. The literature on various types of temperature sensing provides a reference for temperature detection. However, the temperature sensing system based on the Sagnac interferometer occupies an ample space [23] so cannot be well applied in a miniature temperature detection system. Most fiber Bragg grating sensing structures have low sensitivity and are prone to fracture [24], and for traditional cascaded fiber sensing structures, most of the reflective cascade structures require coating technology [25], which is challenging to operate; additionally, the transmission cascade sensing structure has high requirements for the focus position due to the self-imaging effect, so the length of the sensing fiber needs to be strictly controlled [26–31].

In this paper, we propose a miniature MZI temperature sensing scheme based on the SCMCS cascade structure. The transmission Mach–Zehnder interferometer is primarily composed of a single mode–coreless–multimode–coreless–single mode fiber (SCMCS) structure welded together with different core diameters. In terms of temperature measurement, the structure overcomes the dependence of the traditional cascade structure on the length of the sensing fiber. Thus, the length of the sensing fiber has little influence on the sensitivity of the sensor, dramatically reducing the difficult fabrication process for cascade-based structures in micro-sensing schemes. This scheme exhibits characteristics of extensive detection range, small size, good linearity, simple structure, and low cost and has a good development prospect in some detection-related application fields.

## 2. Structure and Principle of the Sensor Head

The SCMCS (single mode fiber–coreless fiber–multimode fiber–coreless fiber–single mode fiber) structure designed in this paper is shown in Figure 1.

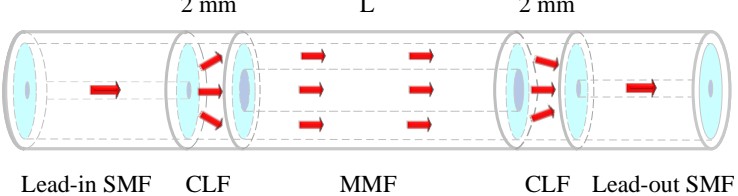

**Figure 1.** Schematic diagram of the SCMCS structure.

Fibers with different core diameters have different patterns and can be explained by Equation (1) [32]

$$V = \frac{2\pi a}{\lambda} \sqrt{n_{core}^2 - n_{cladding}^2} \tag{1}$$

where $V$ represents the normalized frequency of the fiber, $a$ represents the core diameter of the fiber, $n_{core}$ represents the core refractive index, and $n_{cladding}$ represents the cladding refractive index. The higher the normalized frequency is, the more patterns that exist in the fiber. A single-mode fiber has the smallest core $a$, and only the fundamental mode part exists, while a coreless fiber and a multimode fiber have high-order modes in addition to a basic mode, so when welding different fibers, the pattern mismatch leads to the transition of the pattern.

When the light is introduced from the SMF (single-mode fiber) to the CLF (coreless fiber), due to the difference between the core diameter of the SMF and the core diameter of the CLF, the mode is mismatched and the light diffuses in the CLF due to the large core of the CLF; therefore, the light is effectively coupled into core and cladding of the

MMF (multimode fiber). As the refractive index of the fiber core and the cladding of the multimode fiber are different, the optical path difference occurs after output to form a Mach–Zehnder interferometer when the light enters the coreless fiber and couples to the single-mode fiber. When the external physical quantity acts on the multimode fiber and the light passes through the cladding and core of the MMF, an inconsistent response exists, and the interference spectrum changes. (The length of the coreless fiber is 2 mm, the length $L$ of the multimode fiber is 1.5 cm, 2 cm, 2.5 cm, 3 cm, 3.5 cm, or 4 cm, respectively, and the welding machine model is Fujikura 80S, which is produced by Beijing Lingyun Photonics Technology Co., Ltd. in Beijing, China. The single-mode fiber model is G652D, the multimode fiber model is OM2, and the coreless fiber model is CL1010-A.)

Consider the cascading structure as a whole; since it is an interferometric cascade structure, the output light intensity can be explained by the interference theory, that is

$$I = \sum_{i=1}^{n} I_i + 2 \sum_{i=1}^{n-1} \sum_{j=i+1}^{n} \sqrt{I_i I_j} \cos[2\pi(n_i - n_j)L/\lambda] \tag{2}$$

When the light passes through the cladding and core of the MMF, the light interference intensity is obtained as follows

$$I = I_1 + I_2 + 2\sqrt{I_1 I_2} \cos \Delta\varphi \tag{3}$$

The phase difference of the interference light is

$$\Delta\varphi = 2\pi \frac{n_{eff}^{core} - n_{eff}^{clodding}}{\lambda} L = 2\pi \frac{\Delta n_{eff}}{\lambda} L \tag{4}$$

In Equation (4), $\Delta\varphi$ is the phase difference between the cladding light and the core light of the multimode fiber, $n_{eff}^{core}$ and $n_{eff}^{clodding}$ are the effective refractive index of the core and cladding of the multimode fiber, $\lambda$ is the wavelength of the incident light, $L$ is the length of the multimode fiber, and $\Delta n_{eff}$ is the effective refractive index difference between the core and the cladding of the multimode fiber. During the experiment, $m = (1, 2, 3, \ldots)$ level interference troughs were taken to calculate $\Delta\varphi$

$$\Delta\varphi = (2m + 1)\pi \tag{5}$$

Substituting Equation (5) into Equation (4) yields the interference trough wavelength.

$$\lambda = \frac{2}{2m + 1} \Delta n_{eff} L \tag{6}$$

The free spectral range of the interference peaks is

$$FSR = \frac{\lambda_m \cdot \lambda_{m+1}}{\Delta n_{eff} L} \tag{7}$$

$\lambda_m$ and $\lambda_{m+1}$ are the interference trough wavelength of the $m$ and $m + 1$ levels, respectively. At a huge $m$, $\lambda_m = \lambda_{m+1}$, the FSR is inversely related to the fiber length $L$ for the same observation $\lambda$ range and $\Delta n_{eff}$ constant.

When the external temperature changes, thermo-optic and thermal expansion effects cause an optical fiber's refractive index and length to change. According to Equation (6), at this time, the corresponding class $m$ interference trough wavelength varies with temperature and can be expressed as

$$\frac{\Delta\lambda}{\Delta T} = \frac{2}{2m + 1} \left( \frac{\Delta(\Delta n_{eff})}{\Delta T} L + \frac{\Delta L}{\Delta T} \Delta n_{eff} \right) \tag{8}$$

Optimising Equation (8)

$$\frac{\Delta\lambda}{\Delta T} = \frac{2\Delta n_{eff}L}{2m+1}(\nu+\alpha) \tag{9}$$

Both the fiber optic structure thermo-optic coefficient $\nu = \Delta(\Delta n_{eff})/(\Delta n_{eff}\Delta T)$ and the thermal expansion coefficient $\alpha = \Delta L/(L\Delta T)$ are material dependent. Substituting Equation (6) into Equation (9) obtains

$$\frac{\Delta\lambda}{\Delta T} = \lambda(\nu+\alpha) \tag{10}$$

According to Equation (10), when the observation range $\lambda$ is kept constant, the thermo-optic coefficient $\nu$ and the thermal expansion coefficient $\alpha$ are constants; therefore, it is easy to obtain that the temperature sensitivity of the proposed structure will keep constant, which makes the sensitivity of the structure independent of the length of the sensing fiber. Since $\nu$, $\alpha$, and $\lambda$ are all positive, the spectrum will be red-shifted when the temperature increases.

## 3. Experiments and Results

In order to test the performance of the sensing structure, the sensing head was connected to the optical path of a broadband light source, as shown in Figure 2.

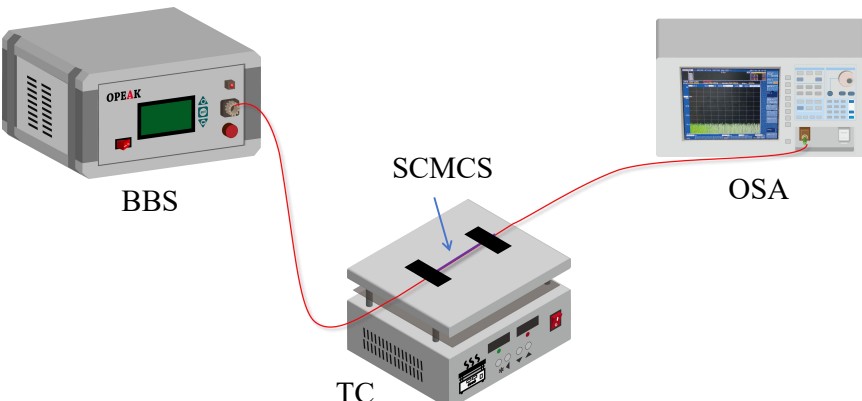

**Figure 2.** Optical fiber temperature sensing system based on a broad bandwidth source.

The Broad Bandwidth Source (BBS) provides a wide spectrum of light, and the temperature controller (TC) provides a constant temperature for the sensing head. The spectrum was observed with an Optical Spectrum Analyzer (OSA), and the sensing structure SCMCS was fixed on the TC. In order to explore the relationship between FSR (Free Spectral Range) and the length of the sensing fiber in SCMCS, the FSR of the SCMCS cascade sensing head structure was observed by changing the length $L$ of the multimode fiber (sensing fiber) at room temperature (the Broad Bandwidth Source is produced by OPEAK OptoElectronics, and the serial number is ASE141201).

As can be seen in Figure 3a, the FSR of the SCMCS structure decreases from 13.6 nm to 4.9 nm as the length $L$ of the multimode fiber increases from 1.5 cm to 4.0 cm. Figure 3b shows the linear fit of FSR and $1/L$. Then, observing the linear fitting plot of FSR and $1/L$ shows that the fit has a good linearity $R^2 = 0.9954$. $Slope = 2.04 \times 10^{-6}$ indicates an effective demonstration of the inverse relationship between FSR and the length $L$ of the multimode fibers. In addition, $Slope = 2.01 \times 10^{-6}$ is theoretically calculated from Equation (4), (Take $\Delta n_{eff} = 0.0125$, $Slope = \lambda_m\lambda_{m+1}/\Delta n_{eff}$, and as shown in Figure 3a, when $L$ = 3.0 cm, then $\lambda_m = 1589.1$ nm, $\lambda_{m+1} = 1582.5$ nm). The theoretical value is very close to the experimental test fitting result, which shows that the experimental results are in good agreement with the theoretical calculation.

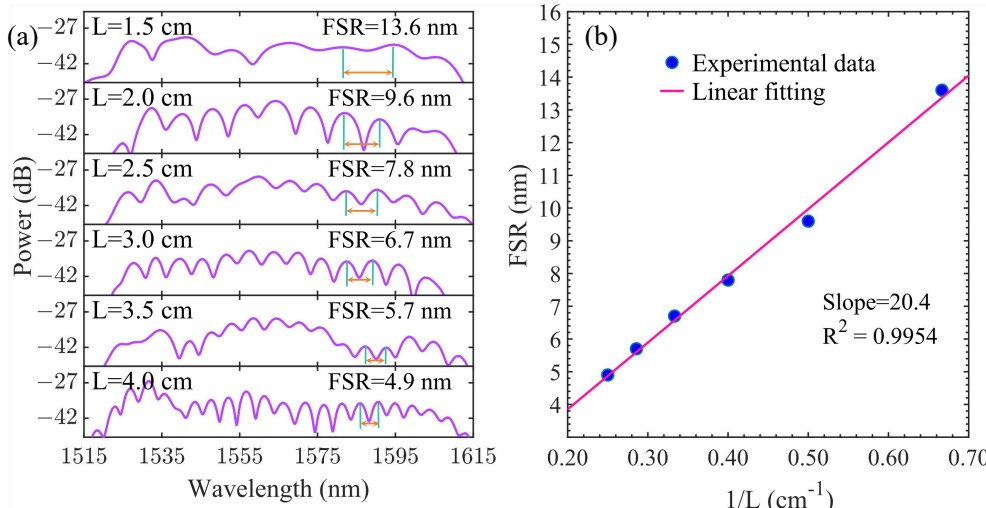

**Figure 3.** (**a**) Spectral shift with different *L* varies. (**b**) Linear fitting of FSR and 1/*L*.

In order to explore the effect of multimode fiber length *L* on sensing sensitivity, the temperature sensing test was carried out on the SCMCS sensor head, which was composed of six groups of multimode fibers with different lengths *L*.

Figure 4 shows the spectral drift of the six groups of SCMCS structures with temperature changes, and the sensing performance of the six groups of SCMCS structures was tested in the range of 20–100 °C (the spectral curve of the starting temperature of 20 °C is the dotted line marked in the figure, the other temperature spectral curves are solid lines, and the trough trend is marked with a black arrow). In Figure 4, (a), (b), and (c) are the spectral drift plots of fiber lengths of 1.5 cm, 2.0 cm, and 2.5 cm, respectively, because the length *L* chosen is relatively short and the corresponding FSR is large; therefore, the experimental phenomenon of wavelength drift can be well observed in the range of 20–100 °C, and there is no overlapping in troughs. (d), (e), and (f) are the spectral drift plots of fiber lengths of 3.0 cm, 3.5 cm, and 4.0 cm, respectively. Due to the limitation of the FSR, the temperature continues to increase and affects the observation; therefore, the temperature range is controlled from 20 °C to 100 °C during the experiment. The trough wavelength data of the six groups of structures with temperature are shown in the Table 1.

**Table 1.** SCMCS structure trough wavelength and temperature data table.

| Temperature (°C) | Trough When *L* = 1.5 cm (nm) | Trough When *L* = 2.0 cm (nm) | Trough When *L* = 2.5 cm (nm) | Trough When *L* = 3.0 cm (nm) | Trough When *L* = 3.5 cm (nm) | Trough When *L* = 4.0 cm (nm) |
|---|---|---|---|---|---|---|
| 20 | 1575.92 | 1577.44 | 1578.55 | 1579.46 | 1572.8 | 1574 |
| 30 | 1576.58 | 1578.08 | 1579.23 | 1580.06 | 1573.44 | 1574.64 |
| 40 | 1577.24 | 1578.72 | 1579.78 | 1580.72 | 1574.16 | 1575.28 |
| 50 | 1577.89 | 1579.32 | 1580.48 | 1581.36 | 1574.84 | 1575.96 |
| 60 | 1578.52 | 1580.08 | 1581.08 | 1581.98 | 1575.56 | 1576.68 |
| 70 | 1579.28 | 1580.76 | 1581.86 | 1582.7 | 1576.36 | 1577.48 |
| 80 | 1579.82 | 1581.6 | 1582.46 | 1583.42 | 1577.12 | 1578.24 |
| 90 | 1580.54 | 1582.4 | 1583.24 | 1584.02 | 1577.92 | 1579 |
| 100 | 1581.21 | 1583.24 | 1584.02 | 1584.8 | 1578.8 | 1579.68 |

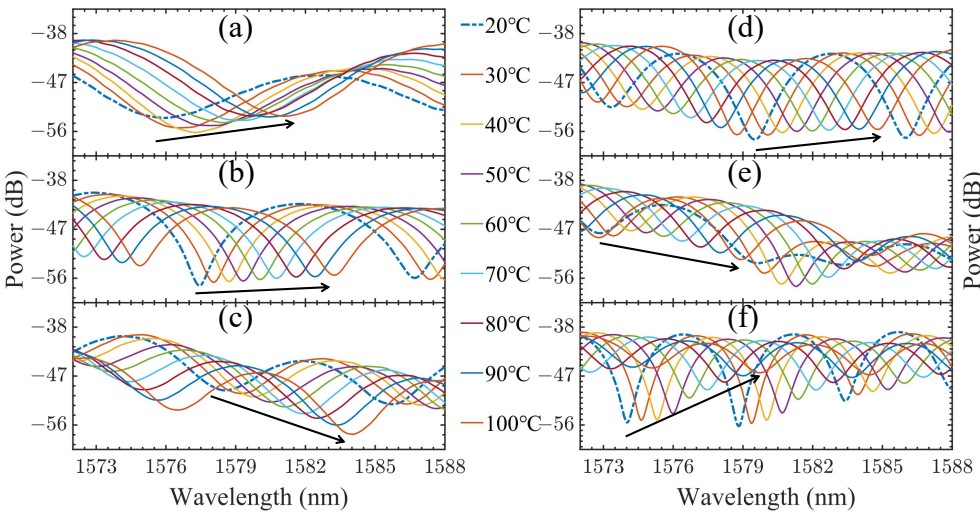

**Figure 4.** Spectral shift with temperature changes. (**a**) *L* = 1.5 cm, (**b**) *L* = 2.0 cm, (**c**) *L* = 2.5 cm, (**d**) *L* = 3.0 cm, (**e**) *L* = 3.5 cm, (**f**) *L* = 4.0 cm.

The wavelength drift of the above six groups of SCMCS sensing structures was linearly fitted with temperature to obtain Figure 5.

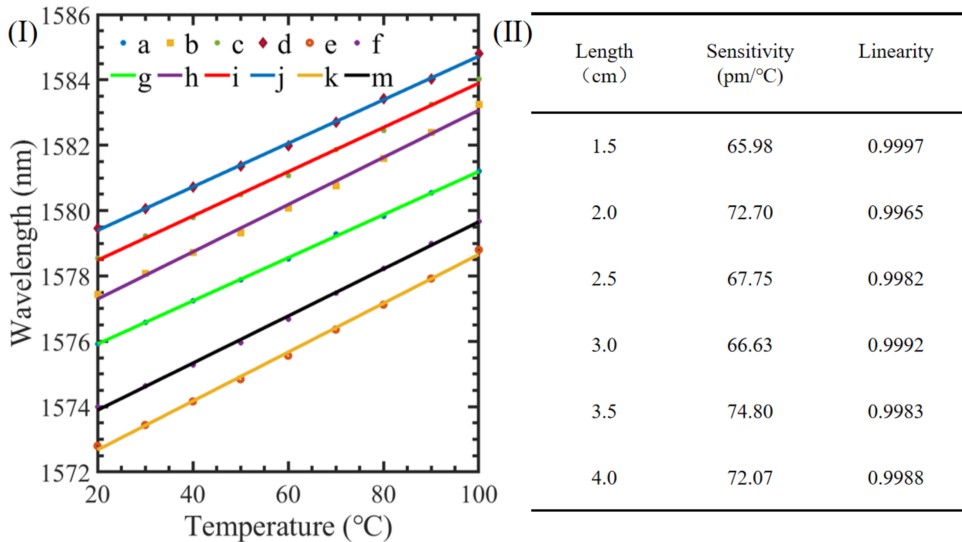

| Length (cm) | Sensitivity (pm/°C) | Linearity |
|---|---|---|
| 1.5 | 65.98 | 0.9997 |
| 2.0 | 72.70 | 0.9965 |
| 2.5 | 67.75 | 0.9982 |
| 3.0 | 66.63 | 0.9992 |
| 3.5 | 74.80 | 0.9983 |
| 4.0 | 72.07 | 0.9988 |

**Figure 5.** Linear fitting of spectral wavelength shift with temperature changes. (**I**) (a–f) Wavelength data for *L* increases from 1.5 cm to 4.0 cm. (g–k,m) The fitting line for (a–f). (**II**) Sensitivity and linearity of linear fitting for structures with different *L*.

The (a–f) of Figure 5I represent the data points of the trough wavelengths for the SCMCS structures with temperature changes when the multimode fiber length *L* varies from 1.5 cm to 4.0 cm, and the (g–k,m) of Figure 5I represent the corresponding linear fitting of the measurement data when *L* varies from 1.5 cm to 4.0 cm. It can be intuitively seen that the fitting lines of each group of SCMCS structures are nearly parallel, confirming that the sensitivity is nearly the same despite the change in the length of the sensing fibers.

Figure 5II shows the sensitivity and linearity values of the linear fitting for each SCMSC structure. By comparing and analyzing, we can see that when the length *L* of the sensing fiber changes from 1.5 cm to 4.0 cm, the sensitivities change little and all of them are positive, indicating that the spectrum is redshifted with an increase in the temperature. These experimental results are in line with the conclusion obtained in Equation (10) that the change in length has little effect on the sensitivity of the SCMCS structure and the wavelength redshift. (Some fluctuations in the sensitivity value are due to errors in the

experimental process, such as poor coupling of the optical fiber welding end face, minor errors in the structure or material of the same optical fiber, etc.)

We have already discussed the variation in sensitivity with sensing fiber length and temperature based on the proposed SCMCS structure in the fixed wavelength range; next, we study the sensitivity of the same fiber structure in different wavelength ranges.

Figure 6 shows the spectrum shift of a structure with $L$ = 2.0 cm as the temperature changes. The wavelength range is from 1530 nm to 1600 nm, and it can be seen that all the drift amplitudes for the different wavelength dips are nearly the same when the temperature changes. Then, we perform a linear fit of the 6 wavelength trough locations to obtain Figure 7.

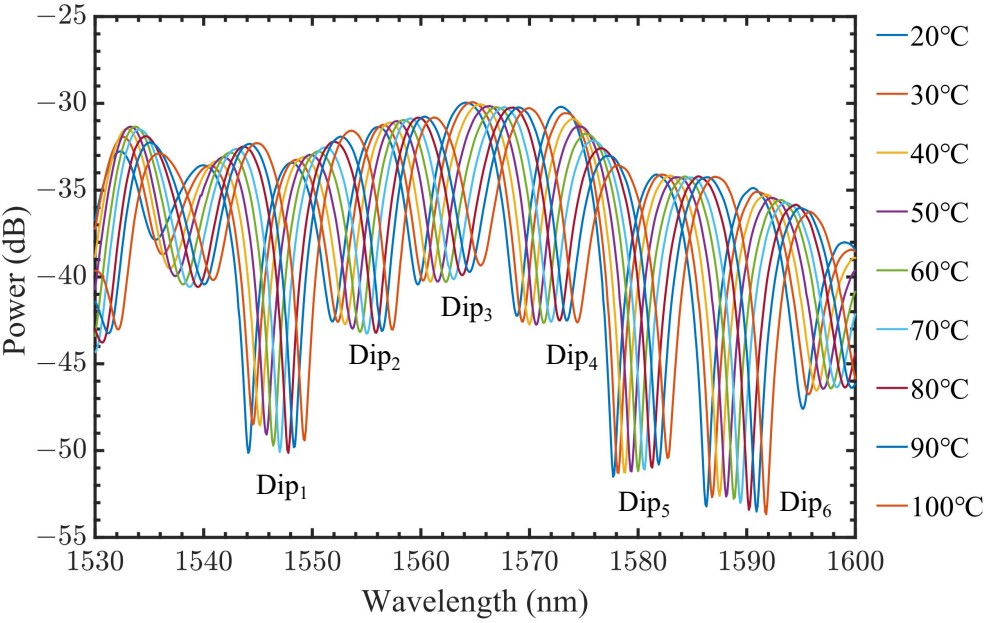

**Figure 6.** The spectrum shift with temperature changes when $L$ = 2.0 cm.

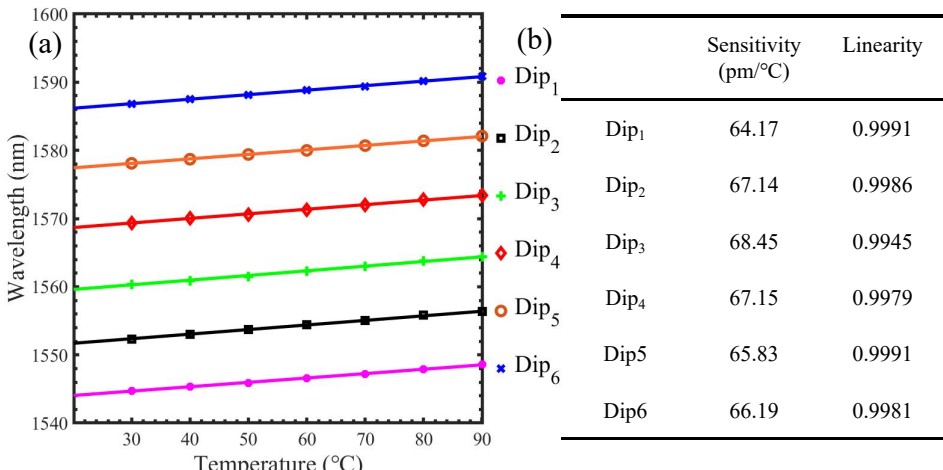

**Figure 7.** Linear fitting of the spectral wavelength shift with temperature changes when the sensing fiber length $L$ = 2.0 cm. (**a**) Diagram of linear fitting for different wavelength dips. (**b**) Sensitivity and linearity of the linear fit at each dip.

Figure 7a represents the linear fit for each point in Figure 6, and the individual fitting lines can be seen to be nearly parallel. Figure 7b shows the linearity and sensitivity of each fitting line, and it can be seen that the linearity of the six groups of fits is above 0.9945, and the temperature sensitivity differences are very small.

In order to study the stability of the proposed SCMCS sensing structure, the spectral variation in the SCMCS structures based on several sensing fibers of different lengths was observed with time at room temperature.

Figure 8a shows the stability analysis data of the trough wavelength of the SCMCS structure of six sensing fibers of different lengths as a function of time. It can be seen that the wavelength fluctuation range of the proposed sensing scheme is within 0.002 nm under different sensing fiber lengths, and the test results prove that the sensing scheme proposed in this paper has good stability and strong repeatability. Figure 8b is the time-dependent variation in the SCMCS structure of $L = 2.0$ cm. The stability of the spectrum over time can be directly observed. Figure 8 shows that the spectrum is stable.

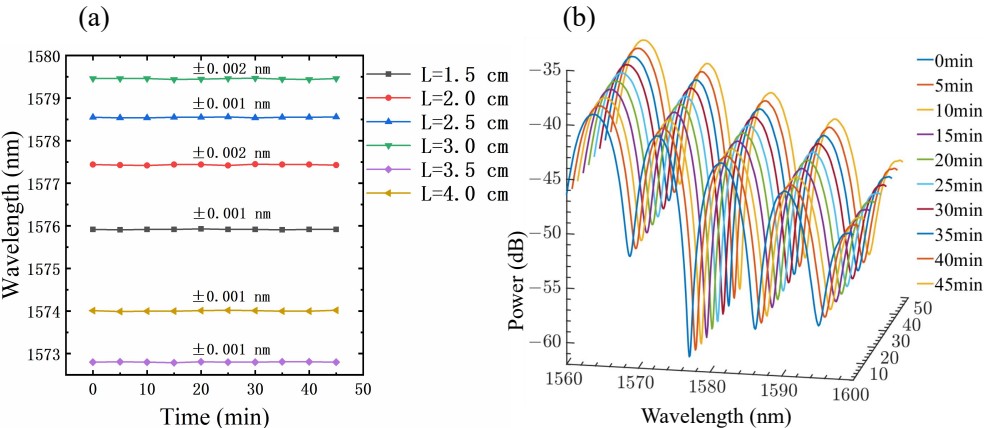

**Figure 8.** Stability diagram of the sensing structures (**a**). SCMCS structures with six different sensing fiber lengths. (**b**) The spectral change for $L = 2.0$ cm.

Table 2 shows some of the temperature sensing schemes proposed in recent years, and their sensing performance is compared with the scheme proposed in this paper.

**Table 2.** Performance comparison of optical fiber temperature sensing schemes.

| Sensing Structure | Temperature (°C) | Sensitivity (pm/°C) | Length (mm) | Reference |
|---|---|---|---|---|
| Triple cladding quartz specialty fiber | 35~95 | 73.34 | 5, 10, 20 | [33] |
| Gallium alloy sensitivity-enhanced FBG | 5~30 | 30 | — | [34] |
| Optical microfiber knot resonator | 27~95 | 14.5 | Diameter = 1.3 | [35] |
| MMF-TCF-NCF-TCF-MMF | 35~90 | 65.24 | 25 | [36] |
| SMF-MMF-SMF | 15~75 | 29.33 | 44 | [37] |
| A thinner no-core fiber | −30~100 | 38.7 | 33.7–35.1 | [22] |
| SMF-NCF | 10~70 | 13.6 | 30 | [38] |
| SMF-HCF-PCF | 25~70 | 10.64 | 1229 | [39] |
| This work | 20~100 | 72 | 20–40 | |

Table 2 compares temperature sensors in recent years in terms of sensitivity, detection range, and length. In fact, there are still many cascades for temperature sensing that are not shown in the table. For example, Li et al. used RCF to weld single-mode fibers at both ends and showed that the sensitivity reaches 64 pm/°C in the temperature range of 30–90 °C [40]. Wu et al. tapered the RCF and welded the single-mode fiber at both ends, and the sensitivity reached more than 171.9 pm/°C in the temperature range of 5–75 °C [41]. Noor et al. used multimode fiber to weld single-mode fibers at both ends, and the sensitivity reached 21 pm/°C in the temperature range of 30–80 °C [27]. These studies were all associated with good results. In this paper, the SCMCS achieved 72 pm/°C temperature sensitivity in the range from 20 °C to 100 °C. It is easy to see that this SCMCS has the advantages of high sensitivity and large sensing range.

## 4. Discussion

Both the theoretical derivation and experiment demonstrate the proposed sensing structure, and the following conclusions can be obtained. Firstly, the sensitivity of SCMCS structures with different sensing fiber lengths in the same observation range is almost the same, and the spectrum drifts forward with temperature. Secondly, the FSR is inversely proportional to the length of the sensing fiber of the SCMCS structure. Finally, the temperature sensitivity differences for the same sensing structure when observed from different positions are very small.

Fiber optic sensors are effective for temperature detection [42,43]. In this paper, it can be seen that the length of the sensing fiber of the SCMCS structure has little influence on the temperature sensitivity, which makes the fiber sensor more convenient in the manufacturing process. SCMCS structures have good sensing sensitivity and linearity coupled with the characteristics of corrosion resistance, electromagnetic resistance, miniaturization, and other characteristics of the optical fiber; therefore, SCMCS structures have a good development prospect, which provides a potential possibility for the development of miniature sensors.

## 5. Conclusions

Based on the principle of inter-mode interference, a temperature sensing scheme based on the SCMCS cascade structure is proposed. The transmission Mach–Zehnder interferometer is primarily composed of a single mode–coreless–multimode–coreless–single mode fiber (SCMCS) structure welded together with different core diameters. In terms of the temperature measurement, the structure overcomes the dependence of traditional cascade-based structures on the length of the sensing fiber, ensuring minimal influence of the sensing fiber's length on system sensitivity. This dramatically reduces the difficulty of the fabrication process for the cascade structure micro-sensing scheme. During the experiment, temperature tests between 20 and 100 °C were conducted with sensing fibers lengths of 1.5 cm, 2.0 cm, 2.5 cm, 3.0 cm, 3.5 cm, and 4.0 cm, respectively. The corresponding sensitivities were 65.98 pm/°C, 72.70 pm/°C, 67.75 pm/°C, 66.63 pm/°C, 74.80 pm/°C, and 72.07 pm/°C, respectively. All corresponding correlation coefficients were above 0.9965. The experimental results demonstrate that the sensitivity of the proposed sensing scheme remains almost constant even under significant changes in the length of the sensing fibers, consistent with theoretical derivation. The sensing scheme exhibits characteristics such as a large detection range, small size, good linearity, simple structure, low cost, and high sensitivity, making it promising for various detection-related application fields.

**Author Contributions:** Q.Y.: Theoretical analysis, Conduct experiments, Data analysis, Article writing; J.T. (Jing Tian): Provide financial support, Theoretical analysis, Revise the article; X.H.: Experimental aids; J.T. (Jiajun Tian): Experimental aids; Q.H.: Software drawing, Revise the article. All authors have read and agree to the published version of the manuscript.

**Funding:** This research was funded by the National Natural Science Foundation of China (Grant number: 61801134, 61835003) and the Guizhou University Audit and Evaluation Project (Grant number: GDSHPG2023007).

**Institutional Review Board Statement:** Not applicable.

**Informed Consent Statement:** Not applicable.

**Data Availability Statement:** Data are contained within the article.

**Conflicts of Interest:** The authors declare no conflicts of interest.

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
