# Peer review of "A Micro-Mach–Zehnder Interferometer Temperature Sensing Design Based on a Single Mode–Coreless–Multimode–Coreless–Single Mode Fiber Cascaded Structure"

_photonics, doi:10.3390/photonics11040363_

Round 1

Reviewer 1 Report

Comments and Suggestions for Authors

Dear Editor,

The authors have presented a SCMCS structure for temperature sensing. The manuscript needs to be polished to reach an acceptance level. So, I recommend that the following comments to be addressed.

1. The manuscript does not present the propagation equations while those are an important section to understand the changes in propagation for sensing operation.

2. What is the criterion for choosing the mentioned values for L parameter?

3. The potentially application of the proposed structure is missed.

4. Do you consider the nonlinear effects in the simulation?

5. The detailed information such as boundary condition, mess cells etc. have not been given.

6. Fiber based sensors such as the following articles have presented for temperature sensing while those have not been considered in the raised scenario in the introduction section.

Twin core photonic crystal fiber for temperature sensing. Materials Today: Proceedings, 33, pp.2289-2292, Optik, 168, pp.342-347, Results in Physics, 16, p.102966

7. what is the advantages of the presented structure than the following works.

Optics Express, 27(23), pp.34247-34257, Optics Express, 27(24), pp.34603-34610

Kind regards,

Reviewer 2 Report

Comments and Suggestions for Authors

The proposed sensor is competitive due to its simple manufacture, low cost, possibility for miniaturization, and good measuring range. This manuscript could be accepted for publication with mandatory revisions.

1. Indicate the product series and company of the BBS broadband fluorescence light source.

2. The authors mention, "In order to explore the relationship between FSR and the length of the sensing fiber in SCMCS, The FSR of the SCMCS cascade sensing head structure was observed by changing the length L of the multimode fiber (sensing fiber)..."

In Figure 3a), Why was the band from 1575 nm to 1595 nm taken to compare the above? The interference fringes are not the same. Wouldn't that affect the comparison? For example, take the band from 1515 nm to 15635 nm.

3. The state-of-the-art only mentions three temperature sensors, one to illustrate each case (Sagnac, Bragg Fiber, and Cascade Type). A critical comparison between the proposed sensor concerning more in cascaded case-like structures is missing, as only two structures are observed in the entire manuscript. In addition, although some comparisons are made in the results section, such as structure, range, and sensitivity, it would be convenient to add the distance of the sensing element from the structure since that is under discussion.

4.- In producing sensors, how do the authors check that the lengths are exactly as proposed with a Fujikura 80S splicer? Do they have any margin for error? How can this affect the MZI signal and sensor production? If greater sensitivity is required, how could authors increase it?

5- Where do the authors see that this configuration can be applied in a real scenario?

Round 2

Reviewer 1 Report

Comments and Suggestions for Authors

Dear Editor,

The manuscript has been enhanced based on the comments so I think the present version can be accepted for publication. 

Kind regards,

Reviewer 2 Report

Comments and Suggestions for Authors

The paper improved with the revision. No further comments